# Redifferentiation of Articular Chondrocytes by Hyperacute Serum and Platelet Rich Plasma in Collagen Type I Hydrogels

**DOI:** 10.3390/ijms20020316

**Published:** 2019-01-14

**Authors:** Vivek Jeyakumar, Eugenia Niculescu-Morzsa, Christoph Bauer, Zsombor Lacza, Stefan Nehrer

**Affiliations:** 1Center for Regenerative Medicine, Danube University Krems, 3500 Krems, Austria; eugenia.niculescu-morzsa@donau-uni.ac.at (E.N.-M.); christoph.bauer@donau-uni.ac.at (C.B.); stefan.nehrer@donau-uni.ac.at (S.N.); 2OrthoSera GmbH, 3500 Krems, Austria; zsombor.lacza@orthosera.com

**Keywords:** articular cartilage, cartilage repair, redifferentiation, collagen hydrogels, biologics, hyperacute serum, platelet-rich plasma

## Abstract

Matrix-assisted autologous chondrocyte transplantation (MACT) for focal articular cartilage defects often fails to produce adequate cartilage-specific extracellular matrix in vitro and upon transplantation results in fibrocartilage due to dedifferentiation during cell expansion. This study aimed to redifferentiate the chondrocytes through supplementation of blood-products, such as hyperacute serum (HAS) and platelet-rich plasma (PRP) in vitro. Dedifferentiated monolayer chondrocytes embedded onto collagen type I hydrogels were redifferentiated through supplementation of 10% HAS or 10% PRP for 14 days in vitro under normoxia (20% O_2_) and hypoxia (4% O_2_). Cell proliferation was increased by supplementing HAS for 14 days (*p* < 0.05) or by interchanging from HAS to PRP during Days 7–14 (*p* < 0.05). Sulfated glycosaminoglycan (sGAG) content was deposited under both HAS, and PRP for 14 days and an interchange during Days 7–14 depleted the sGAG content to a certain extent. PRP enhanced the gene expression of anabolic markers COL2A1 and SOX9 (*p* < 0.05), whereas HAS enhanced COL1A1 production. An interchange led to reduction of COL1A1 and COL2A1 expression marked by increased MMP13 expression (*p* < 0.05). Chondrocytes secreted less IL-6 and more PDGF-BB under PRP for 14 days (*p* < 0.0.5). Hypoxia enhanced TGF-β1 and BMP-2 release in both HAS and PRP. Our study demonstrates a new approach for chondrocyte redifferentiation.

## 1. Introduction

The articular cartilage of the synovial joints, being avascular and limited for endogenous repair capacity, consists of sparsely distributed specialized cells called chondrocytes embedded within its extracellular matrix (ECM), which provide the cartilage with remarkable mechanical and low friction properties. A traumatic injury or anomalous loading to the joint results in a cartilage defect in the short-term. For full articular cartilage defects, cell-scaffold-based tissue engineering approaches, such as the matrix-assisted autologous chondrocyte transplantation (MACT), are some of the standardized treatment methods among many others. MACT involves a two-step surgical intervention by isolating a cartilage biopsy from the non-load bearing site for obtaining and expanding autologous chondrocytes in vitro onto 2D substrates in order to attain sufficient cell populations and then further embedding it onto biological scaffolds and transplantation [1]. The autologous chondrocytes are expanded both under 2D and 3D conditions in the presence of autologous human serum (HS) supplemented along the growth medium [2]. Chondrocyte dedifferentiation is a well-known phenomenon that occurs during 2D expansion, and redifferentiation occurs when cultured on a 3D scaffold [3]. Most often, studies indicating redifferentiation potential on scaffolds utilize fetal calf serum (FCS) and not autologous HS. MACT involves a clinical approach and thereby the use of autologous HS for culturing cell-scaffold constructs. However, supplementing HS does not necessarily redifferentiate in many settings and instead leads to a fibrocartilaginous phenotype usually marked by an increased type I collagen expression with no or less type II collagen expression [4]. Various cell sources other than autologous chondrocytes are investigated for cartilage repair, including bone marrow mesenchymal stem cells, adipose tissue derived mesenchymal stem cells, synovium-derived mesenchymal stem cells and infrapatellar fat pad stem cells. These sources of cells are advantageous for the low donor site morbidity and in obtaining higher cell yield.

HS and platelet-rich plasma (PRP) are blood products. The latter has gained attention over the last two decades for use in cell-culture supplements to expand bone marrow or adipose-derived mesenchymal stem cells (MSCs) [5] and for its advantage over the autologous source. Both autologous and allogeneic pooled PRP have been shown to maintain the stemness and concomitantly support differentiation of MSCs into the osteogenic, chondrogenic and adipogenic lineages. PRP constitutes a milieu of growth factors, chemokines/cytokines, proteases/antiproteases, adhesive proteins, trophic factors, small molecules, and catabolic/anti-catabolic factors [6]. In the clinical context, leukocyte-free PRP is used as intra-articular injections in patients with knee joint degeneration enrolled for randomized controlled trials (RCTs). It provides symptomatic pain relief for up to 12 months, noticed through local adverse inflammation reactions initially after multiple PRP injections, which diminishes over time [7,8,9]. Ex vivo expansion of bone marrow MSCs using autologous PRP has been proposed as and treatment option for articular cartilage defects in a case study with repair, less pain, and mobility but no RCTs have further progressed in this regard [10]. Several in vitro studies investigating the biological effects of leukocyte-free PRP on porcine and human articular chondrocytes observe an increase in both proliferation and differentiation [11,12,13]. Discrepancies among PRP preparations and the mode of activation remains a concern owing to the uncertainty of biological effects and clinical outcomes. A recent alternative derivative of platelet-rich fibrin (PRF) is hyperacute serum (HAS) and HAS which has been consistently reported for its regenerative capacity by increasing the cell proliferation capacity, as reported for chondrocytes [14], bone-marrow mesenchymal stem cells [15], and in an ex vivo model of bone ischemia recently [16]. HAS involves the activation of the natural coagulation cascade by a single-step centrifugation process, and its chemical composition comprises serum proteins, albumin, growth factor, and cytokines. Advantages of HAS over PRP include: there is no cellular reminiscence, it is free from fibrinogen and there is no over-concentration of the plasma content.

An important consideration when using biological scaffolds to regenerate the articular cartilage extracellular matrix is the interaction between chondrocyte and its surrounding niche. In this study, we aimed to test PRP and HAS as an alternative to replace HS for in vitro supplementation in the culture of MACT constructs, namely the collagen type I hydrogel used herein under normoxic/hypoxic conditions. We evaluated the effect of supplementing chondrocytes with HAS and PRP and observed the changes to chondrogenic markers encoding the synthesis of ECM and proteoglycan content. Subsequently as the next approach, we cultured chondrocytes with HAS for seven days and interchanged to PRP during Days 7–14. This way we hypothesized to achieve cell proliferation with HAS and an interchange to PRP would enhance the ECM synthesis. Dedifferentiated chondrocytes were obtained by expansion on 2D substrates for 14 days and subsequently cultured onto collagen hydrogels for another 14 days. Redifferentiation was assessed by analyzing anabolic cartilage matrix gene expression and glycosaminoglycan production as the critical evaluation criteria. Furthermore, the secreted anabolic growth factors and catabolic inflammatory cytokines responsible for modulation of the dedifferentiated or redifferentiation fate were measured in the cell-construct culture supernatant using ELISA.

## 2. Results

### 2.1. HAS Increases Chondrocyte Proliferation and Accumulates sGAG Content but PRP Enhances Anabolic Markers of the Cartilage Extracellular Matrix

The rate of proliferation over the course of chondrocyte redifferentiation analyzed on Days 7 and 14 revealed that PRP inhibited the proliferation over Days 0–14 with no change in cell numbers from the initial seeding density. HAS increased the cell number by three-fold on Day 7 (*p* < 0.0019) and by eight-fold as compared to PRP (*p* < 0.0001) (Figure 1A). DMMB assay quantification of the sulfated glycosaminoglycan (sGAG) illustrated that the total amount of sGAG per construct was enhanced under both HAS and PRP (Figure 1B). The total sGAG content normalized to total dsDNA content denotes an increase in HAS on Day 7 as compared to in PRP (*p* < 0.0121) (Figure 1C).

The redifferentiation potential of the monolayer dedifferentiated chondrocytes embedded in collagen type I hydrogels was assessed for the anabolic/catabolic chondrocyte markers of gene expression. Chondrocytes cultured for 14 days with PRP showed higher expression of COL2A1 on Days 7 (*p* < 0.05) and 14 (*p* < 0.0.5) (Figure 1E) and of SOX9 on Days 7 (*p* = 0.0428) and 14 (*p* = 0.0396) (Figure 1F) than in HAS. Concomitantly, the expression of COL1A1 had a seven-fold increase in HAS during Days 7–14 (*p* < 0.0404), whereas no COL1A1 expression was observed in PRP at both time points (Figure 1D). MMP3 was downregulated as compared to Day 0 in both groups (Figure 1G). MMP13 and VCAN were upregulated in PRP during Days 7–14, whereas, in HAS, MMP13 and VCAN were downregulated during Days 7–14 (Figure 1H,I).

### 2.2. An Interchange in Supplementation from HAS to PRP Enhances Proliferation but Depletes Anabolic Markers of the Cartilage Extracellular Matrix Marked by High Upregulation of MMP13

To achieve both proliferation and redifferentiation of the chondrocytes, an interchange from HAS to PRP during Days 7–14 was performed. The interchange did not arrest cell proliferation (Figure 2A), and the total sGAG content was significantly higher over 14 days (*p* < 0.0238) (Figure 2B). The total sGAG content normalized to total dsDNA content over 7–14 days was decreased by one-fold, denoting a depletion of the sGAG content (Figure 2C).

The anabolic/catabolic chondrocyte markers of gene expression indicated a seven-fold decrease in COL1A1 expression (*p* < 0.05) over 14 days indicating dedifferentiation (Figure 2D). Interchange during Days 7–14 did not enhance the COL2A1 expression (Figure 2E), but SOX9 increased by two-fold (*p* = 0.0952) (Figure 2F). MMP3 was significantly downregulated on Day 14 (*p* < 0.0476) and VCAN was downregulated by three-fold. MMP13 was significantly upregulated by 20-fold (*p* < 0.0238), suggesting that MMP13 upregulation could result in depletion of the extracellular matrix markers and the sGAG content.

### 2.3. Chondrocytes Secrete More Anabolic Growth Factors and Less Catabolic Inflammatory Cytokines When Supplemented by PRP under Normoxia

ELISA quantification of inflammatory cytokine secretion levels under normoxic conditions (20% O_2_) revealed that the chondrocytes secreted higher levels of IL-6 under HAS as compared to the controls; however, in the PRP group, IL-6 secretion was significantly reduced by 10-fold during Days 7–14 (*p* < 0.0446) (Figure 3A). IL-1β levels under PRP were less in the chondrocytes when compared to HAS at 14 days, but no significant difference was observed (Figure 3B). Quantification of anabolic growth factors revealed that PDGF-BB levels secreted by chondrocytes had a two-fold increase in the PRP group during Days 7–14 (*p* < 0.0312) as compared to HAS (Figure 3C). Lower IGF-1 levels were secreted by the chondrocytes than observed in the controls (Figure 3D). BMP-2 was present in a higher amount in both the HAS and PRP controls but the chondrocytes did not secrete BMP-2 under HAS supplementation. Under PRP supplementation, chondrocytes secreted high BMP-2 levels on Day 7, which decreased significantly by Day 14 (*p* < 0.0496) (Figure 3E). No significant differences were observed in the TGF-β1 levels among all groups (Figure 3F).

### 2.4. Secretion of BMP and TGF-β1 by Chondrocytes Is Enhanced by Both HAS and PRP Supplementation under Hypoxia

ELISA quantification of inflammatory cytokine secretion levels under hypoxic conditions (4% O_2_) revealed that the chondrocytes secreted higher levels of IL-6 under HAS as compared to the controls but in the PRP group IL-6 secretion was significantly reduced by seven-fold during Days 7–14 (*p* < 0.0348) (Figure 4A). Lower IL-1β levels under PRP were less secreted by the chondrocytes as compared to HAS at 14 days, but no significant difference was observed (Figure 4B). Quantification of anabolic growth factors revealed that PDGF-BB levels secreted by chondrocytes increased two-fold in the PRP group during Days 7–14 (*p* < 0.0496) as compared to HAS (Figure 4C). Lower IGF-1 levels were secreted by the chondrocytes than observed in the controls (Figure 4D). The above factors were secreted similarly as seen under normoxic conditions. Differences from normoxia were observed in BMP-2 and TGF-β1, where BMP-2 was consistently secreted over 14 days in both PRP and HAS (Figure 4E). Similarly, TGF-β1 was secreted consistently under PRP but decreased over 14 days under HAS. No significant differences were observed between the groups (Figure 4F).

## 3. Discussion

MACT in specific systems utilizes in vitro expansion for engineered cartilage constructs for cartilage repair. We investigated blood products as a natural mixture of bioactive molecules on the chondrocyte microenvironment. Our objective was to assess the redifferentiation potential of dedifferentiated chondrocytes cultured in collagen type I hydrogels for 14 days under supplementation of HAS and PRP in the culture media under normoxia/hypoxia and an interchange from HAS to PRP during Days 7–14. Cell growth was enhanced only by supplementing HAS, and an interchange from HAS to PRP led to enhancement of cell growth. However, the redifferentiation of chondrocytes was achieved only by supplementing PRP for the entire 14-day culture. Proteoglycan accumulation was observed under all conditions, but an interchange led to a decrease in the content at 14 days. Chondrocytes secreted higher pro-inflammatory cytokine IL-6 at seven days under all conditions but reduced at 14 days under PRP. The IL-1β secretion was comparatively lower than IL-6 secretion. The anabolic growth factor secretion of PDGF-BB was higher in PRP at both 7 and 14 days, while IGF-1 remained at same levels under all conditions. Under hypoxia, TGF-β1 and BMP-2 were higher in PRP than in normoxia and vice versa in HAS. The highest magnitude of gene expression encoding cartilage matrix synthesis, namely COL2A1 and SOX9, occurred when supplementing PRP for 14 days, but an interchange from HAS to PRP did not support the synthesis.

MACT involving culture of autologous chondrocytes is often cultured under cell culture medium supplementation of autologous human serum (HS) as a gold standard, but there is not much clarity on the subsequent redifferentiation of monolayer expanded dedifferentiated chondrocytes to MACT constructs for transplantation. Autologous human serum in cell culture supplementation for MACT is known for its progressive proliferation capacity [2], but the MACT constructs very often result in a fibrocartilage tissue formation upon implantation, rather than a hyaline cartilage tissue formation. Discrepancies in analogy with regard to different serum preparations have not been investigated in many of the previous studies articulating MACT constructs. In recent years, several in vitro studies have demonstrated the redifferentiation potential of chondrocytes by platelet derivatives [14,17]. We found different effects on chondrogenesis with different formulations of platelet and serum derivatives wherein our study, PRP inhibited the cellular growth in specific to collagen hydrogels, but matrix turnover markers were enhanced. This mimics the natural environment of chondrocytes as articular cartilage under normal conditions maintains a low matrix turnover and is resilient to proliferation and end-stage differentiation [18]. This result is coherent with several reports that indicate that chondrocytes are involved in the process of proliferation during cell cycle phase or committed to the differentiated state [19,20,21]. Liou et al [22] recently reported that PRP promotes proliferation of mesenchymal stem cells from bone marrow and infrapatellar fat pad in hydrogel encapsulated cultures. This effect could be specific to mesenchymal stem cells and inapplicable to chondrocytes in hydrogels.

Low oxygen tension between 1% and 5% O_2_ is reported to maintain the chondrogenic phenotype during cell expansion and maintain the matrix metabolism in 3D constructs [23,24,25]. However, not all commercially available MACT systems utilize hypoxic conditions for MACT procedures. A few systems, e.g., NeoCart^®^, apply 2% hypoxia in a bioreactor together with hydrostatic pressure stimulation. On the one hand, obtaining clinically sufficient cell numbers without monolayer expansion continues to be a challenge. On the other hand, restoring the functional properties of articular cartilage is insubordinate in tissue-engineered cartilage. To tackle the limitation in increasing cell number while simultaneously enhancing the chondrocytes to resuscitate its matrix synthesis activity in a short culture period within the MACT constructs, an interchange of supplementation of HAS for the proliferation of cells switched to PRP for matrix synthesis was tested in our study. The interchanging of culture conditions, however, did not favor the hypothesis, as proliferation achieving matrix synthesis did not occur, which was indicative of higher MMP13 upregulation after the interchange. The fact that MMP13 is temporarily and not permanently active in articular cartilage as well as its higher activity, based on the congregation of several factors such as insulin-like growth factor (IGF-1), which subsequently results in abnormal homeostasis and breakdown of the proteoglycans [26,27].

Chondrocyte dedifferentiation results from the failure of metabolic imbalance to sustain the anabolic/catabolic equilibrium during synthesis of the ECM involving a multitude of anabolic and catabolic cytokines/growth factors [28]. PRP constitutes a natural milieu of growth factors/cytokines and embodies as an alternative cell culture supplementation to avoid chondrocyte dedifferentiation for MACT procedures. Our study is consistent with other studies where PRP concentrated platelets release the cytokines/growth factors from their α-granules [29,30]. Sundman et al [31] compared the concentrations of anabolic and catabolic growth factors/cytokines in the cellular composition of leukocyte-rich and leukocyte free PRP, and determined that leukocyte-rich PRP (Lr-PRP) released higher levels of catabolic cytokines such as IL-1β, and a higher concentration of platelet content in Lr-PRP released more anabolic growth factors such as TGF-β1 and PDGF-AB. Contrary to the above-mentioned statement, our study involving leukocyte-poor PRP (Lp-PRP) released less IL-1β associated with less TGF-β1 and PDGF-BB. Remarkably, the chondrocytes secreted higher levels of TGF-β1 and PDGF-BB with no increase in IL-1β when supplemented with Lp-PRP. We found an increased secretion of BMP2 by the chondrocytes during the culture period under hypoxic conditions in HAS, which was more in PRP supplementation, as well as the stable release of TFG-β1 throughout the culture time. BMP2 was observed to disappear during monolayer expansion, and the addition of BMP2 was superior to TGF-β1 in preventing chondrocyte dedifferentiation [27]. BMP-2 has been strongly correlated with hypoxia on expression of the matrix gene COL2A1, which is tightly controlled through the p38MAK pathway [32]. However, we did not observe differences in matrix gene expression and the sulfated glycosaminoglycan content between normoxic and hypoxic conditions. This could be attributed to the fact that there might be a prevalence of hypoxia gradients inside the collagen hydrogel. Future investigations should decipher the hypoxia levels and make them more precisely controllable.

This limitation to our current study includes the lack of histological characterization for additional confirmation of the matrix synthesis and remodeling. Deciphering the underlying signaling mechanism between secreted growth factor and cytokines on the ECM turnover would have provided more insights on the effect of PRP and HAS. Taken together, these factors have been considered for future investigations. The current state of the art for cartilage repair envisions on a broad range of tissue engineering strategies and towards this direction a compliant GMP structure can expedite the production of tissue engineered implants and biologics at the premises of the surgical facility. Towards this context additive manufacturing technologies such as 3D bioprinting are advantageous as biomaterials for MACT procedures can be controlled for a high degree of porosity with hierarchical anisotropic architecture for cells to produce collagen fibers vertically, which subsequently helps during the phase of matrix remodeling.

## 4. Materials and Methods

### 4.1. Ethics

The local ethical committee of Lower Austria approved the study protocol on 1st January 2013 (approval No. GS4-EK-4/249-2013). All subjects gave written informed consent in accordance with the Declaration of Helsinki.

### 4.2. Isolation and Culture of OA Chondrocytes

Human osteoarthritic cartilage was obtained from the surgical wastes of 12 donors undergoing total knee arthroplasty (60 ± 3 years old) after written informed consent was given. The cartilage pieces from the superficial zone areas where no cartilage loss occurred were determined then rinsed in phosphate buffered saline (PBS) and minced into fine pieces. Chondrocytes from the articular surface of the cartilage were isolated by enzymatic digestion, as previously reported [4]. Chondrocytes were seeded at a density of 10,000 cells/cm^2^ and expanded in a growth medium (GIBCO^®^ DMEM/F12 GlutaMAX^™-^I, Invitrogen, Vienna, Austria) containing 2.5 µg/mL Amphotericin B and 0.1 mg/mL streptomycin (Sigma, Steinheim, Germany) with 10% FCS (PAA Laboratories GmbH, Linz, Austria). All further experiments were performed on passage 1 chondrocytes to reduce the point of dedifferentiation over passaging.

### 4.3. Preparation of Hyperacute Serum and Platelet-Rich Plasma

Whole blood was collected from 15 individual healthy male and female blood donors (36 ± 10 years old) after written informed consent was given. HAS was prepared by centrifuging whole blood onto 9 mL silicon coated blood collection tubes (VACUETTE^®^ z serum clot activator, Greiner bio-one, Kremsmünster, Austria) at 1770 *g* for 10 min. The top layer containing the supernatant was removed and the resulting fibrin clot (middle layer) was separated from the tube by discarding the bottom part containing red blood cells. The fibrin clot was gently squeezed with a non-absorbable sterile material in a petri dish to extrude HAS. HAS was pooled from the individual 15 blood donors and stored at −80 °C. Leukocyte poor PRP (lpPRP) was prepared by transferring whole blood from the same donors onto 9 mL EDTA coated blood collection tubes (VACUETTE^®^ K3EDTA, Greiner bio-one, Kremsmünster, Austria) and centrifuged at 440 *g* for 10 min. The platelet enriched plasma (middle layer) along the poor platelet plasma (top layer) was further transferred to 15 mL falcon tubes leaving the leukocytes and RBC region and secondary centrifugation at 1770 *g* for 10 min was performed. The resulting lpPRP was pooled from individual donors and stored at −80 °C. Pooled lpPRP enclosed on average 1 × 10^6^ platelets/mL.

### 4.4. Re-Differentiation of OA Chondrocytes by Supplementing Hyper Acute Serum and Platelet-Rich Plasma

Ten milligrams of collagen type I solution (BD Biosciences) were diluted to a final concentration of 2.5 mg/mL in a neutral buffer containing 10× PBS, ultra-pure distilled water, and 1 N NaOH with a final pH of 7.4. Passage 1 dedifferentiated chondrocyte were encapsulated onto collagen type I hydrogels at a density of 40,000 cells/cm^2^ at 4 °C and left to polymerize at 37 °C/5% in a CO_2_ incubator for 30 min. Post-polymerization constructs were re-differentiated in growth medium supplemented with either 10% HAS or 10% PRP for 7 and 14 days under normoxia (20% O_2_) or hypoxia (4% O_2_). Another set of experiments were performed where constructs were redifferentiated in a growth medium supplemented with 10% HAS until 7 days and switched to medium supplementation with 10% PRP post for Days 7–14 under normoxia (20% O_2_) or hypoxia (4% O_2_).

### 4.5. Real-Time Quantitative PCR

Collagen gel-chondrocyte constructs were collected and digested with 120 units/mL collagenase in a serum-free medium for 30 min to release the cells. The RNA was isolated using the High Pure RNA Isolation kit (Roche Diagnostics GmbH, Mannheim, Germany) in accordance with the manufacturer’s instructions. The mRNA was reverse transcripted with a First Strand cDNA Synthesis Kit (Roche Diagnostics GmbH, Mannheim, Germany), and cDNA samples were amplified with RT-qPCR in a cycler. GAPDH was used as an endogenous external reference gene, and the ∆∆Ct method was used to evaluate the relative expression level of mRNA for each target gene. The values are depicted with the Day 0 monolayer dedifferentiated chondrocyte as a control. The following human primers were used in this study: *GAPDH* (forward (F) 5′-CTCTGCTCCTCCTGTTCGAC-3′; reverse (R) 5′-ACGACCAAATCCGTTGACTC-3′), *COL2A1* (F, 5′-GTGTCAGGGCCAGGATGT-3; R, 5′-TCCCAGTGTCACAGACACAGAT-3′), *COL1A1* (F, 5′-GGGATTCCCTGGACCTAAAG-3′; R, 5′-GGAACACCTCGCTCTCCAG-3′), *SOX9* (F, 5′-TACCCGCACTTGCACAAC-3′; R, 5′-TCTCGCTCTCGTTCAGAAGTC-3′), *VCAN* (F, 5′- GCACCTGTGTGCCAGGATA-3′; R, 5′-CAGGGATTAGAGTGACATTCATCA-3′), *MMP3* (F, 5′-CAAAACATATTTCTTTGTAGAGGACAA-3′; R, 5′-TTCAGETATTCGCTTGGGAAA-3′), *MMP13* (F, 5′-TTTCCTCCTGGGCCAAAT-3′; R, 5′-GCAACAAGAAACAAGTTGTAGCC-3′).

### 4.6. Sulfated Glycosaminoglycan (sGAG) and DNA Quantification

Constructs were collected on Days 7 and 14, frozen at −80 °C and lyophilized at 20 °C to evaporate the water content and measure the produced matrix content. sGAG quantification was achieved by treating constructs overnight with 25 U/mL proteinase K enzyme at 56 °C. Enzyme inactivation was then performed at 90 °C for 10 min. One hundred microliters of the supernatant were frozen at −80 °C for DNA quantification and the resultant solution was transferred to ultra-free filter reaction tubes of 0.1 µm pore size (Millipore, Billerica, MA, USA) and centrifuged at 12,000 *g* for 4 min. sGAG was measured through complexation and decomplexation with a 1.9 dimethyl methylene blue solution (DMMB). The absorbance was measured at 656 nm in a plate reader. The DNA content was measured fluorometrically using the Quant-iT™ PicoGreen^®^ assay (Molecular Probes, Vienna, Austria) in accordance with the manufacturer’s instructions (excitation wavelength 480 nm; emission wavelength 528 nm).

### 4.7. Growth Factor and Cytokine Quantification

The above supernatant constructs were collected every 2 days during total media exchange and pooled until Day 7 or Day 14 of incubation as experimental end time points for an enzyme-linked immunosorbent assay (ELISA). The 10% HAS and 10% PRP served as internal controls to the constructs supplemented with 10% HAS and 10% PRP. ELISA kits were used to measure the production of human IGF-I (Quantikine^®^, R&D Systems, Abingdon, UK), PDGF-BB, TGF-β1, BMP-2, FGF-18, IL-1β, and IL-6, and quantified according to the manufacturer’s instructions and measured for absorbance at 450 nm in a microplate reader.

### 4.8. Statistical Analysis

Non-parametric Mann–Whitney two-tailed U-test was performed for comparisons between two datasets at a time. Multiple comparisons were performed using non-parametric Kruskal–Wallis test followed by Dunn’s multiple comparisons test. All data are presented as the mean ± SEM. Significance level was set at *p* < 0.05. All statistical analyses were performed using the GraphPad Prism software (Graphpad Prism Software Inc., San Diego, CA, USA).

## 5. Conclusions

The current study showed that; (i) PRP supplementation over 14 days to collagen-gel chondrocyte constructs enhances the anabolic gene expression markers of cartilage regeneration; (ii) an interchange from HAS to PRP achieves proliferation of chondrocytes but impairs matrix anabolic metabolism; and (iii) hypoxic conditions favor increased secretion of TGF-β1 and BMP-2 by chondrocytes compared to normoxia under both HAS and PRP. Our study proposes a step towards the continual improvement of the existing MACT procedure based on in vitro culture conditions specific to collagen type I hydrogels by instructing the embedded chondrocytes to produce hyaline-like cartilage upon transplantation. Further strategies should be developed for embedding autologous chondrocytes with rapid eTnzymatic digestion post arthroscopy in a PRP-augmented collagen hydrogel, aimed at a one-step cartilage repair procedure.

## Figures and Tables

**Figure 1 ijms-20-00316-f001:**
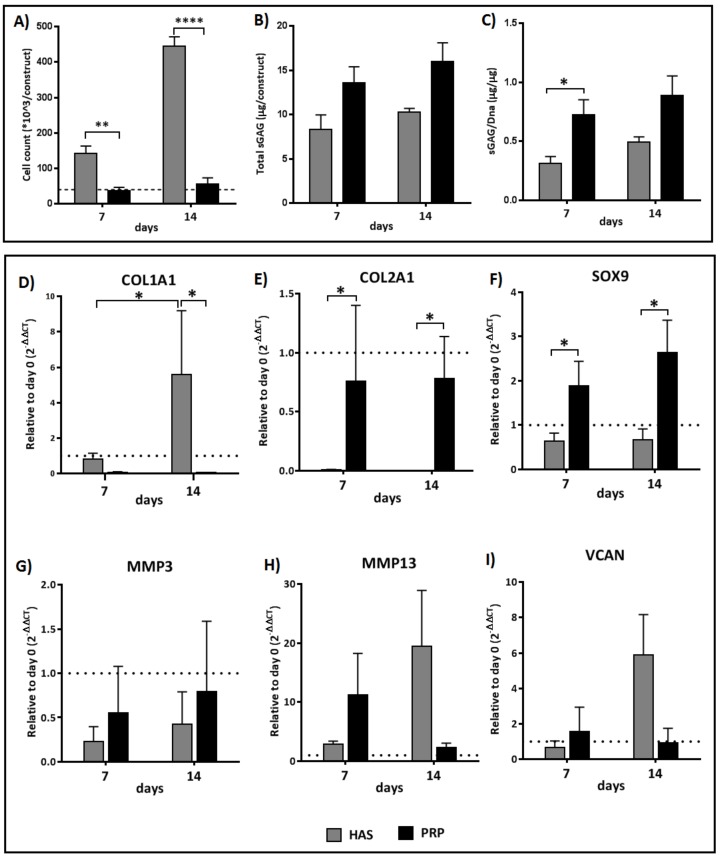
Analysis of: cell proliferation (**A**); total sGAG quantification (**B**); and sGAG/DNA quantification (**C**). Differences in relative expression of chondrogenic markers for COL1A1, COL2A1, SOX9, MMP3, MMP13, and VCAN (**D–I**) as determined by reverse transcriptase quantitative real-time PCR of chondrocytes cultured in hyperacute serum (HAS) and platelet-rich plasma (PRP) under normoxia (20% O_2_). Significant difference at * *p* < 0.05, ** *p* < 0.01, **** *p* < 0.0001; *n* = 6 biological replicates.

**Figure 2 ijms-20-00316-f002:**
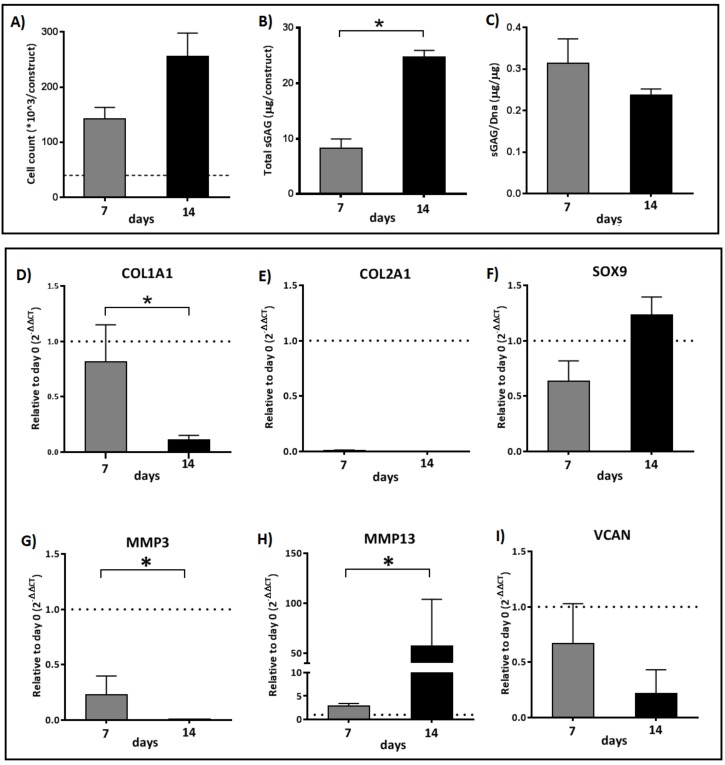
Analysis of: cell proliferation (**A**); total sGAG quantification (**B**); and sGAG/DNA quantification (**C**). Differences in relative expression of chondrogenic markers for COL1A1, COL2A1, SOX9, MMP3, MMP13, and VCAN (**D–I**) as determined by reverse transcriptase quantitative real-time PCR of chondrocytes cultured in hyperacute serum (HAS) for seven days and interchanged to platelet-rich plasma (PRP) during Days 7–14 under normoxia (20% O_2_). Significant difference at * *p* < 0.05; *n* = 6 biological replicates.

**Figure 3 ijms-20-00316-f003:**
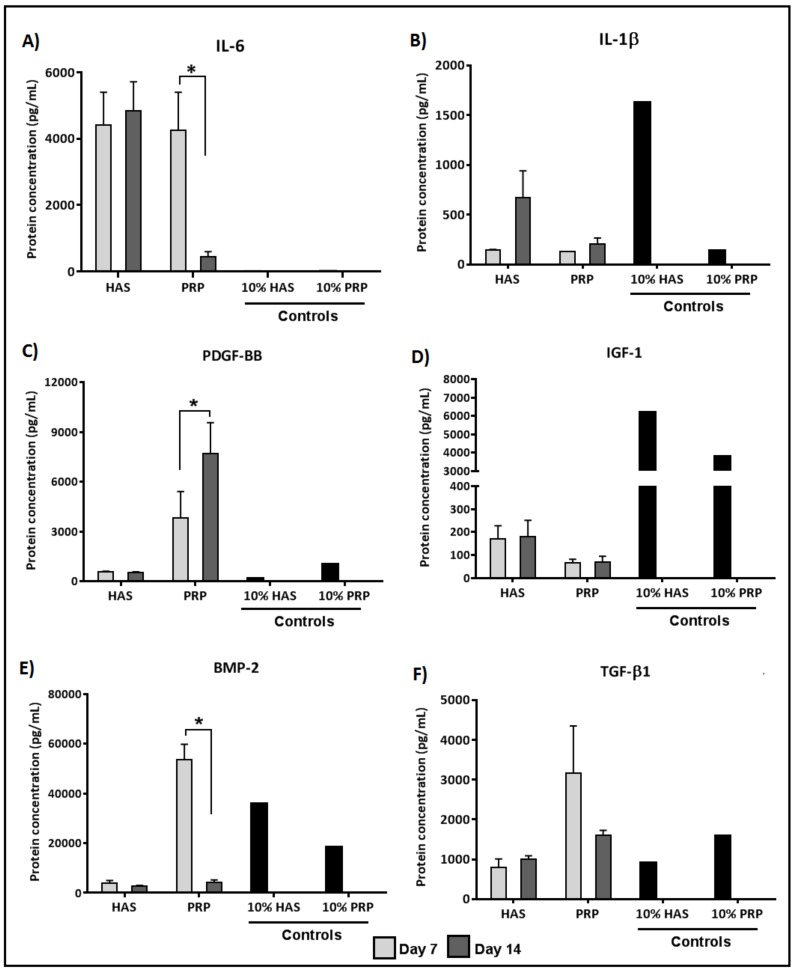
Analysis of determined catabolic inflammatory cytokines (IL-6 (**A**); and IL-1β (**B**)) and anabolic growth factors (PDGF-BB (**C**); IGF-1 (**D**); BMP-2 (**E**); and TGF-β1 (**F**)) secreted by the chondrocytes during the culture period of 7 and 14 days under normoxic conditions (20% O_2_). Significant difference at * *p* < 0.05; *n* = 6 biological replicates.

**Figure 4 ijms-20-00316-f004:**
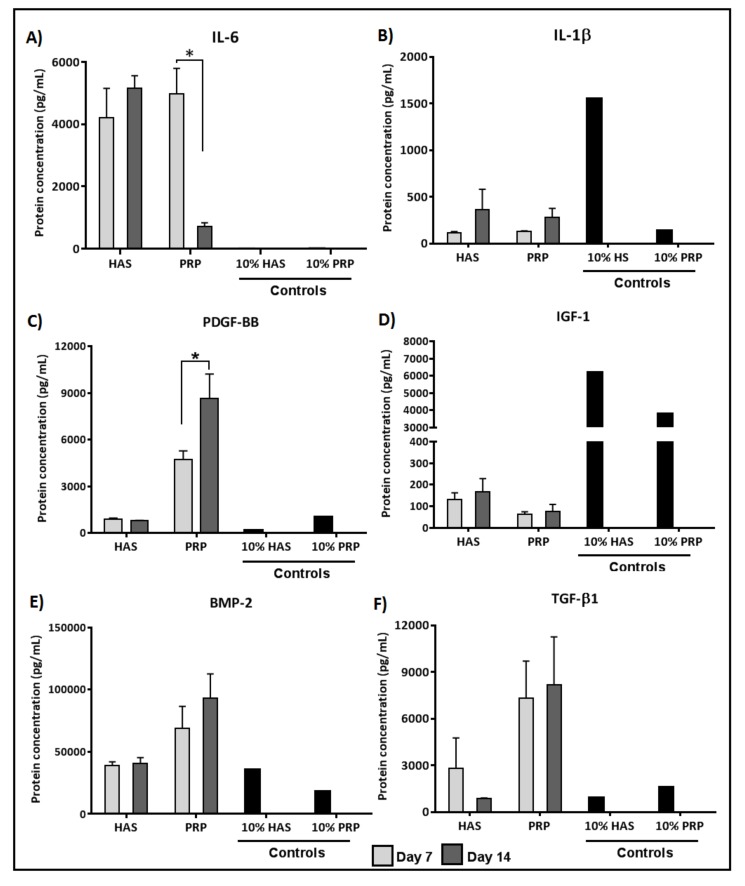
Analysis of determined catabolic inflammatory cytokines (IL-6 (**A**); and IL-1 β (**B**)) and anabolic growth factors (PDGF-BB (**C**); IGF-1 (**D**); BMP-2 (**E**); and TGF- β1 (**F**)) secreted by the chondrocytes during the culture period of 7 and 14 days under hypoxic conditions (4% O_2_). Significant difference at * *p* < 0.05; *n* = 6 biological replicates.

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
