# Peer review of "Redifferentiation of Articular Chondrocytes by Hyperacute Serum and Platelet Rich Plasma in Collagen Type I Hydrogels"

_ijms, 2019, doi:10.3390/ijms20020316_

Round 1

Reviewer 1 Report

Page 1 32-35 This is confusing as written. MACT is generally used for a full articular surface defects.

Page 1 line 40-41: I was not sure how the good manufacturing practice fits into site reference 2.

Page 2 line 53-54: PRP is not an agent for viscosupplementation

Page 2 line 68: Define hyperacute serum (HAS). What are the likely chemical agents in this? The methods show how is it produced.

Page 2 Line 72-76: Not clear what is being said, I think that the only problem is that there are too many concepts covered in two sentences.

Page 2 Line 92: Not sure what the line is saying given that there is no increase in HAS

Page 8 Line 183: When cartilage is submitted for expansion in MCAT what is he usual expansion time in day and what are the standard oxygen conditions?

Page 9 line 244: Precise cell source is hard to tell from the way this is written. Was there an effort to take more superficial chondrocyte from areas no affect by cartilage loss?

Page 9 line 245. Was the PRP preparation leukocyte rich or leukocyte poor? From line 224 appears to be lpPRP.

Author Response

Point 1: Page 1 32-35 This is confusing as written. MACT is generally used for a full articular surface defects.

Response 1: The sentence has been rephrased denoting the usage of MACT for full articular cartilage defects.

Point 2: Page 1 line 40-41: I was not sure how the good manufacturing practice fits into site reference 2.

Response 2: Reference 2 was cited to clarify the use of autologous human serum and the sentence mentioning good manufacturing practice has been deleted.

Point 3: Page 2 line 53-54: PRP is not an agent for viscosupplementation

Response 3: The term viscosupplementation has been deleted and the use of PRP as intra-articular injections had been mentioned herein.

Point 4: Page 2 line 68: Define hyperacute serum (HAS). What are the likely chemical agents in this? The methods show how is it produced.

Response 4: The definition and chemical composition has been added in the introduction section as follows ‘HAS involves the activation of the natural coagulation cascade by a single step centrifugation process, and its chemical composition comprises of serum proteins, albumin, growth factor, and cytokines. An advantage of HAS over PRP is there is no cellular reminiscence, free from fibrinogen and there is no over-concentration of the plasma content.’

Point 5: Page 2 Line 72-76: Not clear what is being said, I think that the only problem is that there are too many concepts covered in two sentences.

Response 5: The sentences have been revised for a better understanding as follows ‘We evaluated the effect of supplementing chondrocytes with HAS, PRP and observed the changes to chondrogenic markers encoding the synthesis of ECM and proteoglycan content. In the next approach, we cultured chondrocytes with HAS for 7 days and interchanged to PRP from 7-14 days. This way we hypothesized to achieve cell proliferation with HAS and an interchange to PRP would enhance the ECM synthesis’

Point 6: Page 2 Line 92: Not sure what the line is saying given that there is no increase in HAS

Response 6: The sentence has been shortened only mentioning the increase in HAS over PRP at 7 days for the sGAG/DNA content.

Point 7: Page 8 Line 183: When cartilage is submitted for expansion in MCAT what is he usual expansion time in day and what are the standard oxygen conditions?

Response 7: General MACT procedures like CaReS® that utilizes collagen type I hydrogels involve a culture period of 14 days pre-implantation. Hence, this time period is adapted in our study. Standard oxygen conditions are 20% normoxia but several commercial systems like Novocart® apply hypoxic conditions to 4% O2. We emphasize future MACT systems to culture the constructs under hypoxia.

Point 8: Page 9 line 244: Precise cell source is hard to tell from the way this is written. Was there an effort to take more superficial chondrocyte from areas no affect by cartilage loss?

Response 8: Thank you for the suggestion; the sentence has been rephrased to ‘Cartilage pieces from the superficial zone areas where no cartilage loss occurred were determined then rinsed in phosphate buffered saline (PBS) and minced into fine pieces.’

Point 9: Page 9 line 245. Was the PRP preparation leukocyte rich or leukocyte poor? From line 224 appears to be lpPRP.

Response 9: Our preparation enclosed leukocyte poor PRP and it has been mentioned in the methods part now.

Reviewer 2 Report

COL1A1 is well known to be a marker of de-differentiated chondrocytes and COL2A1 of differentiated chondrocytes, but in your paper they are both indicated as markers of differentiation. In your resukts, COL1A1 is augmented and COL2A1 decreased in the culture conditions you choose. 

Results are not consistents with discussion, which is too speculative.

Author Response

Point 1: COL1A1 is well known to be a marker of de-differentiated chondrocytes and COL2A1 of differentiated chondrocytes, but in your paper they are both indicated as markers of differentiation. In your resukts, COL1A1 is augmented and COL2A1 decreased in the culture conditions you choose. 

Results are not consistents with discussion, which is too speculative.

Response 1: The authors would like to thank the reviewer for the valuable suggestion. The results have been revised indicating the dedifferentiation status by COL1A1. We agree that our results are speculative but provide a basis for further investigation at the molecular level for signalling mechanisms between the secreted growth factors and their effect on the matrix synthesis. This was not possible to be performed in our current study and is planned to be investigated further. These factors have been now mentioned as a limitation in the discussion.

Round 2

Reviewer 1 Report

New line 203 did not describe standard oxygen tension for cell expansion for MCAT. I thought that is was hypoxic.

Author Response

Point 1: New line 203 did not describe standard oxygen tension for cell expansion for MCAT. I thought that is was hypoxic

Response 1: A new sentence (Line 378) has been added indicating the standard oxygen tension conditions for MACT ‘Low oxygen tension between 1% O2 to 5% O2 is reported to maintain the chondrogenic phenotype during cell expansion and maintain the matrix metabolism in 3D constructs [23], [24], [25]. However, not all commercially available MACT systems utilize hypoxic conditions for MACT procedures except a few systems like NeoCart® apply 2% hypoxia in a bioreactor together with hydrostatic pressure stimulation.’

The authors would like to mention that the English language style has been revised by the proofreading services from Papertrue English editing services.

Author Response

The authors would like to mention that the English language style has been revised by the proofreading services from Papertrue English editing services.